# Spatiotemporal Data Mining Problems and Methods

**Eleftheria Koutsaki, George Vardakis and Nikolaos Papadakis \***

Department of Electrical and Computer Engineering, Hellenic Mediterranean University, 71410 Heraklion, Greece
\* Correspondence: npapadak@cs.hmu.gr

**Abstract:** Many scientific fields show great interest in the extraction and processing of spatiotemporal data, such as medicine with an emphasis on epidemiology and neurology, geology, social sciences, meteorology, and a great interest is also observed in the study of transport. Spatiotemporal data differ significantly from spatial data, since spatiotemporal data refer to measurements, which take into account both the place and the time in which they are received, with their respective characteristics, while spatial data refer to and describe information related only to place. The innovation brought about by spatiotemporal data mining has caused a revolution in many scientific fields, and this is because through it we can now provide solutions and answers to complex problems, as well as provide useful and valuable predictions, through predictive learning. However, combining time and place in data mining presents significant challenges and difficulties that must be overcome. Spatiotemporal data mining and analysis is a relatively new approach to data mining which has been studied more systematically in the last decade. The purpose of this article is to provide a good introduction to spatiotemporal data, and through this detailed description, we attempt to introduce descriptive logic and gain a complete knowledge of these data. We aim to introduce a new way of describing them, aiming for future studies, by combining the expressions that arise by type of data, using descriptive logic, with new expressions, that can be derived, to describe future states of objects and environments with great precision, providing accurate predictions. In order to highlight the value of spatiotemporal data, we proceed to give a brief description of ST data in the introduction. We describe the relevant work carried out to date, the types of spatiotemporal (ST) data, their properties and the transformations that can be made between them, attempting, to a small extent, to introduce constraints and rules using descriptive logic, introducing descriptive logic into spatiotemporal data by type, when initially presenting the ST data. The data snapshots by species and similarities between the cases are then described. We describe methods, introducing clustering, dynamic ST clusters, predictive learning, pattern mining frequency, and pattern emergence, and problems such as anomaly detection, identifying time points of changes in the behavior of the observed object, and development of relationships between them. We describe the application of ST data in various fields today, as well as the future work. We finally conclude with our conclusions, with the representation and study of spatiotemporal data can, in combination with other properties which accompany all natural phenomena, through their appropriate processing, lead to safe conclusions regarding the study of problems, and also with great precision in the extraction of predictions by accurately determining future states of an environment or an object. Thus, the importance of ST data makes them particularly valuable today in various scientific fields, and their extraction is a particularly demanding challenge for the future.

**Keywords:** data mining; spatiotemporal data; spatial data; description logic; geographical data; time series; raster; point



## 1. Introduction

Spatial data are data derived from spatial measurements that have only one parameter: the space and its characteristics are mainly location coordinates and constitute the mining data. Spatial data mining leads to the relationship between these data, derived from their

combination and interdependence, to draw conclusions about a location. Spatiotemporal data, however, are data derived from measurements, which take into account both the parameters of space and time. Spatiotemporal data mining leads to the extraction of clustering relationships and modeling of the ST data in order to extract patterns. During spatiotemporal mining, patterns emerge, leading to predictive conclusions. For example, studying and recording weather changes over a period of time can lead to deriving a weather pattern and predicting weather changes in the future [1]. The comparison of spatial and spatiotemporal data mining can be seen in Table 1.

**Table 1.** Comparison of spatial and spatiotemporal data mining [2].

| Features | Spatial Data Mining | Spatiotemporal Data Mining |
|---|---|---|
| Definition | Extraction of information and relationships from geographical data stored in a spatial database | Extraction of information from the spatiotemporal data to identify the pattern of the data |
| Data | Needs space information within the data such as location coordinates, etc. | Needs space and time information |
| Conceptual basis, Rules | Based on rules like association rules, discriminant rules, characteristic rules, etc. | Based on finding patterns in the data by clustering, association, prediction and data comparison |
| Application | Determining the hotspot of any event | Register situation changes over a period of time and predicts a future state |

Various sciences, such as medicine, meteorology, and social sciences, can process the data extracted at a specific location and time to draw important conclusions and make important observations in various fields [3]. Spatiotemporal (ST) data are constantly changing over time. There are several ways to extract such data today to extract useful information, taking into account both the space and the time at which they are collected.

What makes spatiotemporal data unique compared to classical data is the dependence they show on both place and time at the same time. The traditional method of data mining assumes that the measurements are not affected in time, that they are independent of each other, and that they are identical; a hypothesis that cannot be valid when extracting spatiotemporal data, since when measuring ST data, the snapshots taken are directly related to the time taken in a particular area, and their properties and characteristics are directly affected by the temporal and local position of the measurement. If this dependence is not taken into account when extracting the data, then we are led to incorrect and inaccurate results.

STDM takes snapshots that can have different characteristics depending on the method used. Spatial measurements, for example, take snapshots/objects that acquire characteristics depending on the time at which they were taken. Thus, each site is now an object with different characteristics derived from measurements at different times in the same area, or a group of sites can be considered as an object with characteristics derived from measurements over a period of time. A typical example in meteorology is the study of different regions that show similar climatic phenomena over time. In this case, the areas with similar climatic phenomena are grouped and are objects in which specific characteristics are determined that are measured at regular intervals. Another approach to the same problem is to take the different time points as objects and the variables that will determine their characteristics to come from measurements performed at different locations. A typical example of temporal objects is the extraction of patterns of human brain activity from neuroimaging where the object is the time at which a particular brain activity takes place, by determining the characteristics of the object [4–6]. An object defined by time and place could be an event/action. An example of an object, which refers to an action, is a crime, as it takes place at a specific place and time. This helps to draw conclusions

about the frequency and where a particular crime is being committed. When determining characteristics, in addition to the time and space, taking into account the nature and extent of the crime and the interrelationship of all these characteristics, is today a very demanding challenge in terms of extraction and analysis of space-time data. The dependence of the measurements based on the time and place where they are made creates the need for new methods of extraction and analysis of space-time data (spatiotemporal data mining/STDM) or a combination of already existing methods.

This article tries to classify the types of spatiotemporal data, which solve problems of a specific form and category, with their parallel grouping, based on the methods used to solve them. It also categorizes data mining methods that aim at grouping, extracting patterns, predictive learning, and identifying anomalies, changes, and relationships between data, as each category of problems is researched in terms of the types of ST data it uses.

## 2. Related Work

Extensive research and study has been carried out in the extraction and analysis of spatiotemporal data. In recent years, efforts have focused on extracting time series data and extracting spatiotemporal data for problems such as trajectory identification and spatiotemporal grouping of objects.

The scope of such an approach is in medicine, where human brain activity is represented by neuroimaging data, which record both the point in the brain where the action is observed and at what time it occurs [5–7]. Google adapts time to its maps by analogy, data that also change over time. Spatiotemporal data today is perhaps the answer to many questions in various scientific fields.

In the age of big data, STDM algorithms are applied in various scientific fields such as meteorology, social media, and medicine, with special requirements in computing power [3]. Using STDM, groupings, predictions, and visualizations can be performed in various scientific fields. Attempts have already been made to export and analyze orbital data, as well as to create ST types and standards from a database. Due to the wide variety of problems that STDM has to deal with regarding a large number of applications, this has created the need to determine the structure and scope of research of STDM. Identifying the types of problems and methods as well as the differences between them is a fundamental need in the field of STDM. This will yield valuable knowledge because we will now be able to clearly distinguish, which solution of which problem that its structure can be a solution, and for someone else, or for a category of other problems. For example, the method for extracting climate patterns can also be a method for extracting patterns of brain activity [8].

Spatiotemporal data are not a conceptual dataset but are grouped into various types according to their properties. Below, we present in detail the types of spatiotemporal (ST) data with their respective characteristics, we compare them based on the type to which they belong, and we present how one type of ST data can convert to another type and be used for solving another type of problem.

## 3. Spatiotemporal (ST) Data

SUMMARY OF Section 3

In this section we describe the types of ST data, specifying constraints and rules using description logic (Section 3.1), the properties of ST data (Section 3.2), and the conversions that can be made from one type to another (Section 3.3); finally (Section 3.4), we describe the resulting snapshots per type (Section 3.4.1) and the similarities between them (Section 3.4.2). INTRODUCTION OF Section 3

The ST data are of great interest today, so a study is made in terms of how to extract them, but also in terms of how they are used in various applications. Depending on the purpose of their use, the ST data are divided into different types and have specific properties, with each type having distinct differences from the others. Each type of ST data results in a snapshot with a specific structure and properties that are entered and analyzed in an application.

Thus, spatiotemporal data are classified into various types, depending on the properties they present.

*3.1. Types of ST Data*

Depending on the application, which uses space-time data, the type of ST data is also determined. The values of ST data differ depending on the time and place, where they are collected each time, on the types of applications they serve, on the types of problems that they called to solve, and on the way they are extracted which is different each time. Thus, in order to be able to make the right choice of data mining method, it is necessary to first determine the application for which they are intended and then determine the type of data collected. ST data are divided into four types: event data, trajectory data, reference point, and raster data. Event data refers to data that describe an event at a specific time and place, for example, the study of a crime. Traffic data are those that describe the orbit followed by a moving object. Reference point data are a set of values, collected from specific points in an area. Finally, raster data are data collected from measurements made at a set of values in a specific time and local context, for example, the study of brain activity using FMRI (functional magnetic resonance imaging). The properties of each type of ST data are determined based on the application that is going to use that data. An ST data type can have completely different properties each time it is used depending on the purpose for which the data were collected and the application they are going to serve, and, vice versa, a set of ST data, which has a specific type and serves an application, can change type and serve a completely different application. For example, a type of ST orbital data can be converted to an ST raster type. Thus, taking advantage of an ST data extraction method (STDM method), we can utilize these data in different applications, each time giving a different type to them. Let us now look at each type of ST data in detail and identify limitations and rules per type ST data, using description logic.
EVENT DATA

An event is determined by a space-time value, which indicates the time and place in which an event took place. For example, a criminal act is considered as an event which is accompanied by a space-time value that determines the time and place in which it was committed. For example, a car (a) theft is an event that took action at a certain time $t_I$ and place $l_i$: TheftCar($t_I,l_I,a$), (Example-1).
Limitations:

(1) Two different objects, $a_1$, $a_2$, are not allowed to be at the same time $t_1$, at the same time point $l_1$. For example, two different cars, $a_1$, $a_2$, are not allowed to be at the same time $t_1$, at the same point $l_1$, and to declare, for the same time and space, the theft of both of these cars. Thus, TheftCar($t_1,l_1,a_1$) $\neq$ TheftCar($t_1,l_1,a_1$).
(2) An object is not allowed to be presented in two different places at the same time. Position($t_1,l_1,a_1$) $\neq$ Position($t_1,l_2,a_1$).
(3) Can an object appear in the same place at two different times, only if a strictly defined time interval is defined? For example, Position($t_1,l_1,a_1$) = Potion($t2,l_1,a_1$) when $0 \leq t \leq 100$ s, so it is immovable. When t does not belong to the above interval, then it means that, probably, the object moved to another point, $l_2$.

Based on these constraints we can proceed to define rules for the event data. Thus,

(1) Each object has a unique position in space for a given moment in time.
(2) Two objects have a unique and different position at a given moment in time.
(3) An object can be considered to be at the same point at two different times, $t_1$, $t_2$, when a time interval t is defined where $t_1 \leq t$ and $t_2 \leq t$, where in this case the object is considered stationary in this period of time.

Using descriptive logic, the above rules are expressed as follows:

(1) TheftCar($t_i$, $l_i$) $\rightarrow \neg$ Position($a_1$, $t_i$, $l_i$) $\wedge$ Position($a_2$, $t_i$, $l_i$) $\wedge$ $a_1 \neq a_2$.
(2) Move($a_1$, $t_1$, $l_1$) $\wedge \neg$ move($a_1$, $t_2$, $l_2$) $\rightarrow$ Position($a_1$, $t_1$, $l_1$).
(3) Move($a_1$, $t_1$, $l_1$) $\wedge < t \wedge \neg$ move($a_1$, $t_1$, $l_2$) $\rightarrow$ position($a_1$, $t_1$, $l_1$).

Here, t is a time interval, a is a car, and $l_1$ and $t_1$ are specific locale and time points.

Similarly, an infectious disease is considered to be an event with a spatiotemporal value at the time and place that a person first became ill. A set of events is characterized as a spatial point model by using a set of ST values [9]. In a Euclidean coordinate system, a spatial point is determined by the values of $l_i$ and $t_i$, which are its spatial and temporal values, respectively. An event in addition to time and place may have other types of information, such as the nature of an event, e.g., of a crime, such as theft, information that constitutes the labeled variables. A spatial point that represents an event, e.g., a crime, depending on the value of a labeled variable presented, can take three forms: A-circle (e.g., murder), B-square (e.g., robbery), or C-triangle (e.g., theft) (Figure 1). Event data are widely used today in many fields, such as medicine, meteorology, social media, transportation, etc. Sometimes events are represented in a Euclidean coordinate system as points, but there are cases in which we are more interested in another dimension, for example, a car accident in addition to the time it takes place and point on the road axis that it takes place. We are also interested in its distance from other traffic accidents that occurred at another point on the road axis; in this case, the event is not represented as a point in the Euclidean coordinate system, but as a polygon or a line [10]. An example of an event in the form of a polygon is the identification of a burned area as a result of fire. In addition, an event is not always characterized by a specific time value, which determines its occurrence value, but can be an entire time period with beginning and end as the start time and end time of the event.

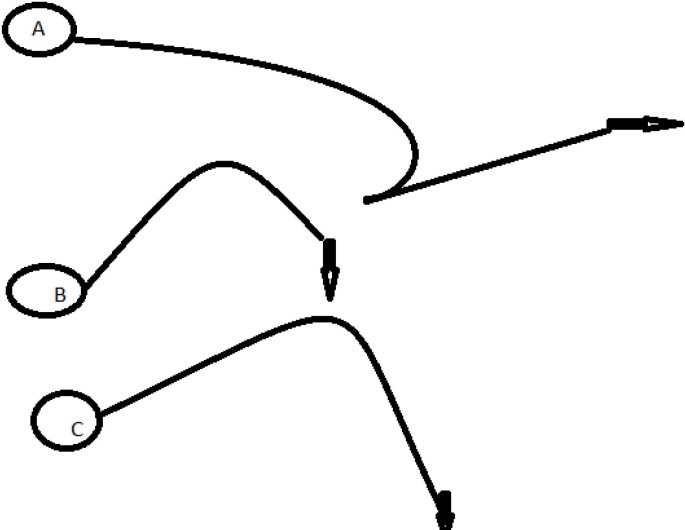

**Figure 1.** Events belonging to three types: A—cycles, B—squares, and C—trajectories. Trajectories of three moving objects A, B, and C. Events and trajectories data types.

For example, the occurrence of a fire in a forest area is an event that is represented, using the start time of the fire and the end time, where the seizure was made. If a fire occurs at a spatial point $l_i$ at a specific time $t_i$, then the set of all neighboring spatial and temporal regions, which represent the event of fire, in a specific temporal and spatial extent, constitutes the set of points forming a polygon, depicting the extent of the fire:

$$\text{ExtentofFire}(t,l) = \sum_{ti,li}^{ta,\ lb} \text{Fire}(t,l), \quad \text{(Example-2)}.$$

**Limitations**

(1) A polygon formed by a number of neighboring spatial points ($l_1$, $l_2$, $l_3$, $l_4$), with a set of events (ev1, ev2, ev3, ev4), respectively, cannot, with the number of these neighboring points, form another polygon at the same time ($t_1$) and determine its extent of the same event. So it can be formulated as follows:

$$\text{Polig}_1(l_{1ev1}, l_{2ev2}, l_{3ev3}, l_{4ev4}) \neq \text{Polig}_2(l_{1ev1}, l_{2ev2}, l_{3ev3}, l_{4ev4}) \text{ for a } t_1.$$

(2) Two polygons may consist of the same spatial points at the same time when they describe different events. Thus,

$$\text{Polig}_1( \, l_{1ev11}, l_{2ev21}, l_{3ev31}, l_{4ev41}) = (l_{1ev12}, l_{2ev22}, l_{3ev32}, l_{4ev42})$$
$$\text{where ev11} \neq \text{ev12, ev21} \neq \text{ev22, ev31} \neq \text{ev32 and en41} \neq \text{ev42}$$

**Rules**

(1) When we study the spatial extent of an event, at a specific moment in time, then the resulting polygon is unique.
(2) Two or more polygons may be the same when, for the same spatial and temporal points, they studied a different event.

Using description logic, the above rules are expressed as follows:

(1) $\text{Polig}_1(l_{1ev11}, l_{2ev21}, l_{3ev31}, l_{4ev41}, t_1) \wedge t_1 \equiv t \rightarrow \neg \, \text{Polig}_2 (l_{1ev12}, l_{2ev22}, l_{3ev32}, l_{4ev42}) \wedge t_2 \equiv t_1$, where t is specific *period* of time.
(2) $\text{Polig}_1 (\forall l_{ev}, t_1) = \text{Polig}_2 (\forall l/_{ev}, t_2)$, if $t_1 \equiv t_2 \wedge \forall l_{ev} \neq \forall l/_{ev}$.

ORBIT DATA

The path followed by an object at a specific time and place is represented in the form of an orbit. One route could be the route of a taxi, starting at a passenger pickup point and ending at a disembarkation point. For the study of moving bodies, a sensor is placed on them, which records both the time and the spatial point in which they are located. Figure 1 shows the orbits of bodies A, B, and C. An orbit can, in addition to time and space, also have labeled variables. An example of a labeled variable could be the speed at which a taxi travels during a journey. The trajectories find grounds of application in various fields such as transport, criminology, etc.

Each trajectory path is determined by the following function $f(t) = \alpha t^2 + \beta t + \gamma$, where $\alpha$, $\beta$, and $\gamma$ are the highlighted variables/coefficients that can additionally characterize a trajectory of a moving object, such as the inclination of the trajectory.

For example, the path taken by a taxi starting from Athens airport (li), at 2 p.m. until 5 p.m. at peak time, is as follows: $\text{LocationofTaxi}(t) = \alpha t^2$.

Another labeled variable for a trajectory could be the speed ($u = s/t$) of the object (taxi). Thus, by using the speed we can know at any moment in time where the object is. For example, at time $t_1$ the object/taxi will be at the point.

$l_1 = u * t_1$ (see Figure 1, trajectory C (Example-3).

Limitations

(1) Two objects cannot have the same trajectory when moving at the same speed in the same amount of time.
(2) Two trajectories are not considered similar when all their points are not exactly the same.
(3) A trajectory does not consist of one point over a period of time.

Rules

(1) Two objects can delete the same trajectory as long as they have different speeds for different amounts of time or same time period but different start time. For example, the faster object will clear the trajectory in a shorter time.
(2) Two trajectories are considered similar when all their points are the same.
(3) To be considered a trajectory, a trajectory must have more than one spatial point in a defined time interval.

Using *description logic*, the above rules are expressed as follows:

(1) $\text{Traj}(a_1, t_{start1}, u_1, l_{start1}) \wedge \neg \text{Traj}(a_2, t_{start1}, u_2, l_{start2}) \wedge u_1 \equiv u_2 \wedge l_{start1} \equiv l_{start2}$, where $t_{start}$ and $l_{start}$ are the temporal and spatial starting points.
(2) $\text{Traj}(a_1, t_i, u_1, l_i) \wedge \text{Traj}(a_2, t_i, u_2, l_i) \wedge (u_1 \neq u_2 \vee t_1 \neq t_2)$.
(3) $\text{Traj}(a_1, t_1, l_i) \wedge \neg \text{Traj}(a_2, t_1, l_j) \wedge \exists l_i \neq \exists l_j$.
(4) $\text{Traj}(a_1, t_1, l_i) \wedge \neg i \equiv 1$.

POINT REFERENCE DATA

The values obtained from the measurements of a continuously changing variable, such as temperature and population, in relation to time and place, are the data of reference points. The population of an area, for example, is constantly changing over time. Thus, a reference point in relation to the population is the value that the population has, in a specific area and in a specific time point. A past set of ST values /reference points is also the context in which a space-time field moves. From the study of two or more space-time fields, another space-time field can be extracted, using various methods, for any other time period, (as shown in the paper "Atluri, Gowtham, Anuj Karpatne, and Vipin Kumar. "Spatio-temporal data mining: A survey of problems and methods." ACM Computing Surveys (CSUR) 51.4 (2018): 1–41.", Figure 3 [11], that shows a space-time field, (with circles the reference points) of an area at two different time values, where different aspects of ST grids used to represent raster data. All sites in the ST raster can either be located regularly or irregularly in the space. The set of time stamps can also be either normal or irregular).

For example, on 22 September 2020 ($t_1$), the population of Athens ($l_1$) was 4,000,000; this value is a reference point. On 22 September 2021 (t2), the population of the same city was 4,250,000, and in 22 September 2022 (t3), the population was 4,280,000. The set of these reference points constitutes a set of ST reference points, which forms a spatiotemporal field of values and which testifies to the growth rate of the population of Athens in the last three years (Example-4).

Limitations

We cannot qualify as reference points, for a point in time, measurements that are not carried out at regular time intervals at the specific point.

Rules

(1) In order to consider two or more spatial points as reference points, the times during which measurements were made at these points should have a constant difference between them, i.e., $t_2 = t_1 + \alpha$ and $t_3 = t_2 + \alpha$, where $\alpha \in R$.

(2) A reference point can also be a time point. For example, at a time $t_1$ we measure the temperature in three large cities of Greece, for example, on 22 September, the temperature in Athens was 28 °C, in Heraklion 30 °C, and in Thessaloniki 26 °C. The reference point in this case is the time, so 22 September can capture data as a timestamp.

Using description logic, the above rules are expressed as follows:

(1) Measures($t_i$,$l_i$)$\wedge$ti $\equiv t_{i-1}$ + a, *where a $\epsilon$ R.*

(2) Measures($t_1$,$l_i$)$\wedge$li $\equiv l_{i-1}$ + a, *where a $\epsilon$ R.*

RASTER DATA

Raster data are the set of values that are recorded when studying a field at fixed positions both in space and time. Initially, fixed points in space S = $s_1$, $s_2$, ..., $s_n$ are selected, which can be evenly distributed in space.

The measurements each time for each point of the space are made in a specific set of finite values of time T = $t_1$, $t_2$,..., $t_n$. The measurement times can be consecutive or even have a constant difference between them.

The values are recorded in a Cartesian system with S and T axes, where each edge in the ST grid that is created is also a measurement.

Raster data are widely used in the study of various scientific fields such as medicine and the study of climate change. A typical example is the measurement of temperature in an area or group of areas at regular time points. A specific example of raster data is the temperature measurements that can be carried out in the summer months in three large cities of Greece: Athens, Heraklion, and Thessaloniki. On 23 June: 33 °C in Athens, Heraklion 30 °C, Thessaloniki 31 °C; on 23 July: Athens 35 °C, Heraklion 33 °C, and Thessaloniki 32 °C; on 23 August: Athens 38 °C, Heraklion 35 °C, and Thessaloniki 34 °C (Example-5).

Limitations

(1) Measurements taken in an area at unspecified times cannot be considered raster data.
(2) Measurements made in a set of different areas for a given moment in time are not considered raster data.

Rules

(1) Raster data are considered the measurements made at a spatial point $l_1$, at regular time intervals, $t_1$, $t_2$, $t_3$, where $t_2 = t_1 + a$ and $t_3 = t_2 + a$, with $a \in R$.
(2) Raster data can also be considered the measurements made at a given time $t_1$, in a group of areas, as long as these areas are related to each other in terms of some characteristics (for example, they are spatially adjacent).

Using description logic, the above rules are expressed as follows:

(1) $\forall \text{Measures}(t_i, l_1) \wedge t_i \equiv t_{i-1} + a$, *where a $\epsilon$ R*.
(2) $\text{Measures}(t_1, l_i) \wedge l_i \equiv l_{i-1} + a$, *where a $\epsilon$ R*.

In addition to the measurement of brain activity in a specific brain area at regular time points, using the FMRI methods [12], a raster point could also be a value derived from a set of measurements from sensors, performed at a point peak for each cell in the grid. An example of an ST data raster might be the population value on an annual scale with measurements performed in cities or regions. Of great interest are the conclusions that can be drawn from the analysis of raster ST data. An example is in the field of climate change: valuable conclusions, about the climates changes, about their frequency and periodicity, are drawn from the measurements on the Earth's surface using Landsat instruments with a spatial resolution of 30 m with measurements every 16 days or with the use of measuring instruments with coverage 500 m and taking measurements every day [13]. Finally, there is the possibility of greater analysis or more general analysis in data raster's, depending on the problem they are called to solve.

For example, for a larger analysis of the space or time of a given raster in Geographic Information Systems (GIS) the sampling method is used [14]. Conversely, for various reasons, such as due to limited computing power or to avoid duplication during measurements, raster ST data can be represented by appropriate groupings and as collections of cells. In this case, however, the possibility of losing valuable information during the analysis should be taken very seriously.

However, every type of spatiotemporal data is distinguished by some properties. These properties are related either to the heterogeneity or to the dependence between the data, always in relation to the space and time in which they are collected.

*3.2. Properties of ST Data*

The main properties of ST data are two: the self-correlation and heterogeneity of data. The correlation of ST data with each other usually occurs when the data are collected at close times and/or spatial distances. When this happens, it is observed that the data are "related" to each other. The reason for this relationship is due to the spatial choice of data collection, for example, the selection of nearby areas for measuring traffic, or due to time selection of data collection, for example, at very close times for which the temperature is measured in an area. Thus, when the ST data to be studied depend directly on both space and time of collection, they cannot be collected to be analyzed and processed by classical algorithms, as this would lead to erroneous and inaccurate results. This therefore leads to the need to develop algorithms for data mining, which take into account both space and time.

However, there are also spatiotemporal data that are independent of each other. This heterogeneity is due to the different ways, times, places, and levels in which they are collected. For example, in the study of climate change, based on the measurement of temperature, changes are presented as cyclical time patterns. As a result, the temperature values recorded, for example, in summer are distributed differently from those in winter. Another typical example is the study of brain stimulation in a stimulus, each stimulus

activating a different area of the brain. It is therefore observed that there can be heterogeneity in the study cases either in relation to time or space and this leads to the need to develop distinct models of mining methods, depending on the type of ST data and the type of problems they are called to solve [15].

However, each type of ST data can be converted to another data type in order to serve another application and solve a different type of problem.

### 3.3. Data Types Conversion

The ST data collected by a specific application belongs to a specific type, but the type can be changed and the same data can be used for other analyses with similar STDM tools. For example, ST event type data can be converted to ST raster type data, grouped into a set of events, in the form of a grid cell. For example, temperature measurements, which are benchmark data, at specific points in an area can be measured on an hourly basis and converted to a raster temperature measurement ST.

For example, the temperature in Athens on 22 September at 14:00 was 17, at 13:00 it was 18, at 14:00 it was 20, and at 15:00 it was 19. This gives the ST raster: Athens (17,18, 20, 21) with t E (12:00 until 15:00) on 22 September, where each temperature measurement in Athens constitutes a given reference point (Example-6).

Vice versa, a type of ST raster data can be converted to other types of data, for example, when locating areas of criminal activity using appropriate methods, where a raster is the value of crime in a city and with an appropriate method, we extract the crimes that occur in an area as ST event data. Consequently, reference point data can also be extracted from a raster, assuming that each grid top ST is also a reference point ST.

Whether or not the type of ST data is converted, from each final type of data to be processed, a snapshot with a specific structure and properties is obtained that is imported and analyzed in an application.

### 3.4. Data Instances

3.4.1. Data Instances Per Type

The snapshot is the input data of an ST data mining application (STDM). In an STDM algorithm, the snapshot consists of a set of measurements of the properties of an object to be studied, depending on the type of data we conclude and for different types of snapshots. There are five different ways of representing the instances for STDM applications, each for every type of ST data type. Each data type can be represented by different types of snapshots, as shown in Table 2.

**Table 2.** ST data of various types can represent various classes of ST data instances.

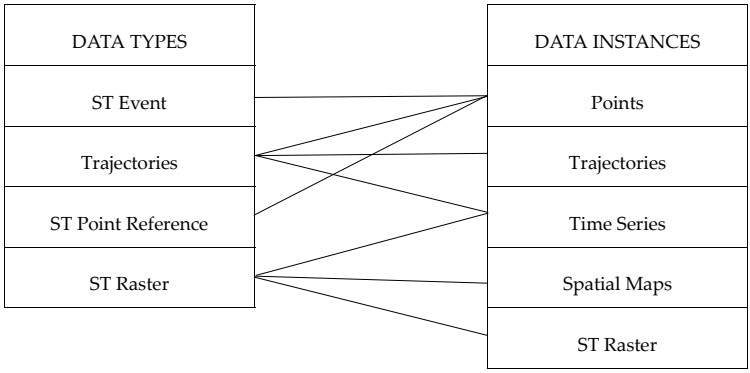

For example, event type data can be represented by point type snapshots while a trajectory can be resisted both by point type snapshots and trajectory type snapshots or even as a spatial time series. The coordinates of each given type of reference point are represented as a point snapshot in the field. For raster data there are three ways to represent snapshots. The first way is to define a snapshot as the set of measurements at one location

within a time domain; this will result in a time series snapshot (look at Example-6). The second way is to consider a snapshot as the set of measurements at any given time in a set of locations, and this will result in spatial map type instances (look at Example-5). Finally, an instance could be considered the whole grid, depending on the problem that is being called to solve with the appropriate application, the corresponding extraction of a specific type of data, and the corresponding conversion into a specific type of instances, in order to satisfy the needs and requirements of an algorithm each time. However, let us look in more detail at each category of instance.

Points Instances

A set of measurements of an attribute, performed as a function of time and space, represent a snapshot data ST point. STDM applications often use points instances, which represent instances and refer to events. A snapshot of an event point could be a specific event, for example, a crime, that occurs at different locations and times. This can lead to the grouping of such events (crimes) that show similarities or even to the analysis of differences between events that do not follow the form of most, for example, theft of private cars of a certain brand in different areas of Athens (Example-7). A point snapshot can also be considered as a snapshot of reference points, from measurements, to estimate a space with unknown cases and then to be represented in instances. In addition, a snapshot orbit can be broken down into snapshots of points at which an object moves. When representing instances, very interesting conclusions can be drawn about their interaction. For example, if a collection of point instances follows a pattern, if we can distinguish instances of instances that deviate from the majority of the set, or if we can track the value of a variable that has not been studied during mining. An example of a function that attempts to highlight the correlation of instances is Ripley's function k.

Trajectories Instances

Another type of ST data instance is the orbit, which studies the motion of an object. A snapshot of orbit is a collection of spatial points that are collected at different times, sorted into a list. Useful conclusions that can be drawn when studying snapshots are the grouping of similar tracks and the creation of groups of tracks, where each group describes a different form of route in addition to the extraction of continuous points of an orbit followed by other moving objects, i.e., the extraction of intersection points between trajectories.

For example, a study and comparison between the most frequent routes followed by Athens taxis in the summer months was used in order to draw conclusions such as the most popular places for tourists (Example-8).

Time Series Instances

Time series snapshots can only be extracted from two types of ST data: the raster and the trajectory. Initially in raster data we can assume that the data represent measurements, which refer to a specific spatial grid and are performed at different times, so a raster can be considered as a time series snapshot (look at Example-6). ST orbit type data, in turn, can be interpreted as measurements of a variable point in space over time, that is, as time series snapshots with two-dimensional data time and space or more, which describe the course of an object. Using time series snapshots, an orbit can be represented by a set of spatial points, independent of each other, that is not limited to a frame, something that opens new horizons in the study of orbits.

When studying time series snapshots, valuable conclusions can be drawn, such as grouping time series with similar activity in a specific time frame, exporting a pattern between a set of time series, exporting time stamps or short intervals from a set of time series where the expected behavior from the time series is followed, the prediction is extracted after analyzing a set of time series, and the prediction for a future time series, studying the older successive time series (look at Example-8).

Spatial Maps Instances

Spatial maps are ST instances of raster data collected at one point in time from a spatial value field [16]. Conclusions that can be drawn when analyzing spatial maps are as follows: exporting a group of spatial maps with similar behavior at different times, exporting patterns between spatial maps, predicting a value in a variable from studying a series of spatial maps, and the prediction of the value of a variable in a location by observing its values in other locations at a given time.

In Example-5, where the temperature measurements for the three cities can be taken in all three summer months, we can draw safe conclusions about what temperatures prevail in the three major cities of Greece during the summer months by drawing similarities for the variations of temperature.

Raster Instances

Raster ST data are the large ST chapter in ST data analysis. Raster data are very often used as input data in the form of instances in various applications. Conclusions that can be drawn when studying raster snapshots are as follows: the grouping of raster's in sets which have similar behavior in time and space, the export of raster snapshot patterns, the extraction of raster snapshot cases that do not follow the general form of a set of snapshots raster, the export of time points at which a change in the format of the raster snapshot is observed, the prediction of the value of a variable target when analyzing raster snapshots, and the export of parts of the grid where an interesting action is observed. The representation of ST rasters is performed with N-way arrays, the tensioners. A raster is represented by a tensor using two or more dimensions: one or more to represent the spatial points as a whole and the other to represent the set of time points in a grid. Tensor examples are the raster representation of ST data relating to the FMRI method for the study of brain activity, where the data have four dimensions: three for space and one for time [17]. From the study of a set of tensioners with similar values, a tensor can be extracted with specific values (these values or the partial formatting of a tensor with similar values); this method is known in image segmentation [17,18]. ST rasters are displayed in grid formats in order to extract their characteristics. Each node is a spatial point and each edge of the network represents the similarity between time series. Among the nodes, depending on the characteristics that a node receives, their role also changes. Thus, for example, one network may have nodes with properties, related to social characteristics, and the same network may have nodes with biological characteristics or nodes with properties related to a transmission network [19].

For example, the study of the type of crime in 2022 in areas of Athens can be carried out by extracting a network with nodes representing the areas of Athens with values like the level of crime at 2022 and edges like the time distances between crime measurements. The same network can change form and functionality, at the same nodes and have measurements for the economic situation of the residents, with results now drawing different conclusions (Example-9).

A great advantage of temporal data over other forms of data is the similarities between their instances. The comparison and similarity that arise in some cases between instances belonging to a type can lead to the derivation of complex and valuable conclusions.

3.4.2. Similarity between Cases Instances

The answer to many STDM data mining problems comes from comparing STDM instances. The similarity between the instances leads to grouping, to the identification of patterns, and to the general extraction of relations between them. Vice versa, the inhomogeneity between the instances leads to interesting conclusions, such as the detection of instances that do not follow the general form of a set. The similarity or the heterogeneity are observed in all types of ST instances.

Similarities per type instances:

POINTS INSTANCES

Identical points are considered points that belong to the same neighborhood. A neighborhood of a points is considered a set of points with similar characteristics that belong to a frame with a specific space (fixed distance of spatial points) in a certain period of time. An example of a neighborhood is the set of points belonging to a distance of one meter in the space of one hour. A neighborhood can also be considered a set of finite points that are relatively close to each other (without determining the exact and constant distance between them). Thus, one point is considered similar or close to another when both belong to the same neighborhood.

ORBIT INSTANCES

Similarity between two orbits is observed when, over time, moving objects are very often close to each other in their path. Orbits may have similarities as a whole or they may have similarities only in specific parts of them. Today, the similarities in track subsequences, similarities in the longer part of the route, etc., are studied. Of interest in the study of orbits are the patterns that emerge from the observation of similar orbits to draw useful conclusions, such as which route most tourists choose at a particular time of day in a tourist destination.

TIME SERIES INSTANCES

Assuming that a time series is a one-dimensional representation of data with one dimension, the temporal, then the comparison between time series can be made, either by calculating the time distances between the observations they record, or by comparing the correlations for the data of each time series, or even comparing the patterns or shapes extracted from each time series. Another measure of comparison could be the time delay between the observations, represented by each time series between the recording of observations, with the final choice as time series, which represents the greatest similarity in the time delay intervals (choosing, for example, the one with the longest lag). Two time series do not always show similarity throughout their duration, but, on the contrary, there is a greater or lesser correlation between them in some of their time intervals, where, in this case, patterns are exported that concern only certain time intervals and not the entire duration of the time series.

SPATIAL MAPS INSTANCES

The similarity between spatial maps instances is realized by comparing the values, which are recorded as points on the paper. Many times, however, this method leads to wrong conclusions because the geographical locations on the maps have not been determined correctly. To avoid such errors, we compare sub regions of the maps with each other instead of studying the similarities of the whole map [20,21].

RASTER INSTANCES

Raster snapshot similarities can be obtained by comparing the networks that each snapshot of rasters represents. The comparison is made in terms of the number of nodes and vapors that the networks have, but also in terms of the final characteristics/conclusions that are extracted, from the representation of ST data, in the form of an instance network.

However, in order to be able to compare the ST data and, consequently, to group them, both the time and the space in which they are collected should be taken into serious consideration during this process. This poses a challenge and creates many problems when applying the appropriate method in trying to collect, compare, and group the appropriate data aimed at solving a problem.

## 4. Problems and Methods

SUMMARY OF Section 4

In this section we describe the groupings that can be performed by type of ST data (Section 4.1), the detection of dynamic groupings by type (Section 4.2), the stage of predictive learning by type (Section 4.3), the frequency and appearance of pattern mining (Section 4.4), the anomalies presented by the ST data (Section 4.5), the identification of time

points of change in the behavior of the objects to be studied (Section 4.6), and, finally, we describe the correlations between time series (Section 4.7).

*4.1. Clustering*

Based on the comparison of values obtained by the properties of ST data in order to find similarities between them, their grouping is performed. However, in the case of ST data, things are more complicated than other types of data, because when comparing them, both the time and space parameters must be taken into account [22,23].
ST POINTS

There are two ways in which clustering ST points are interesting. The first way is to cluster the points based on the number of points that appears in one cluster, and the second way is to compare one cluster against another.
POINTS BASED ON THE NUMBER OF POINTS REPRESENTING CLUSTERS OF POINTS

The goal is to find two or more clusters with an unusually high number of points. These clusters are also called hotspots. Hotspots are of interest in various fields such as detecting the onset of infectious diseases or human gatherings or even detecting the occurrence of an unusual activity on social media, also known as the "incident detection" problem [3]. The study for finding hotspots has been extensively studied in classical spatial statistics [24]. Nowadays, however, a great deal of research is being carried out on the study of ST data in terms of identifying hotspots, which will indicate either an outbreak of an infectious disease or an outbreak of criminal activity. This study of ST data is also called "space-time scanning statistics".
POINTS BASED ON THE BEHAVIOR OF A CLUSTER OF POINTS

Locating a set of points that have the same values in case characteristics is another criterion for grouping point clusters, for example, the grouping of areas where the same criminal activity is observed in the same time periods. The ST-DBSCAN algorithm is one of the algorithms that performs this type of clustering; it is an evolution of DBSCAN that clusters classical spatial data [25,26]. In the ST-DBSCAN algorithm, both spatial and temporal distances between points are determined by the user, taking into account other nonspatial properties, thus offering great flexibility in customizing targeted queries [10,26].
TRAJECTORIES

The clustering of the trajectories is based on the comparison of the trajectories that delete moving objects. Thus, for example, we can locate taxis that delete similar routes or the route followed by two hurricanes [27].

One method of grouping tracks can be by modeling mixtures [9,28]. Another method of grouping can be to join two points/tracks of the moving object via GPS, to extract the path, (as shown in the paper "Atluri, Gowtham, Anuj Karpatne, and Vipin Kumar. "Spatio-temporal data mining: A survey of problems and methods." ACM Computing Surveys (CSUR) 51.4 (2018): 1–41.", Figure 5 [11], where mobility profiles discovered by a user's GPS tracking based on travel grouping (grouping tracks A and B from the paths/lines method of grouping can be by joining two points / tracks of the moving object via GPS, to extract the path) [29]. However, there is often interest in grouping track sections. In this case, the TRACLUS method is followed, where the tracks are divided into sections that are compared and grouped with each other through the DBSCAN method [26]. Of additional interest is the finding of "moving clusters" as a form of clustering in which a set of moving objects, whether leaving or integrating into a cluster, such as a moving cluster, is the movement of migratory birds or car processions.
TIME SERIES

Time series from ST rasters, which consist of the time recording of measurements in space, are grouped based on the grouping of continuous areas that are represented by the same frequency of time activity. This problem is addressed by various methods such as the K-means grouping method, the hierarchical grouping, the grouping of the common nearest neighbor, etc. for example, a related research has been carried out where time series are grouped, based on sea surface pressure measurements across the land surface over

a period of 11 years, from 1982 to 1993, where the comparison is based on the Pearson coefficient [30–32].

Conventional time series grouping methods do not guarantee cluster extraction in contiguous regions; however, this is overcome by increasing the number of clusters or removing clusters with discontinuous locations from a set of clusters. Another approach is to place a cluster on the basis of proximity. With the "developing area" method, adjacent areas that have many similarities are merged, creating a single cluster of clusters, with the ultimate goal of having the lowest possible number of different clusters [33].

SPATIAL MAPS

The grouping of spatial maps is carried out by locating time points where there are similarities in the spatial maps, which are ST rasters, from measurements at specific time intervals in specific areas. An example of grouping spatial maps is the grouping of brain areas with similar brain activity at a given time [34]. This grouping extracts continuous time zones with similar activity into a cluster of spatial maps. Many times, however, what is required in a problem is the identification of continuous time sets. This is performed when smaller clusters of maps are analyzed with distant time stamps between them, so as to lead to clustering of different clusters between them.

### 4.2. Finding Dynamic ST Clusters

One difficulty that occurs during grouping is the identification of subareas, both in space and in the time period of ST rasters, where consistent values are presented in terms of measurements. These areas are also known as "dynamic complexes". The determination of subregions presents the existence of phenomena that modify a part of an area during a period of time. Lakes, for example, are a dynamic ST complex of the total aquatic world, as over time they change spatially (grow or shrink), changing in size, shape, or overall appearance [35]. Consequently, a cluster of ST sites can be considered dynamic, as it is modified over time. A dynamic ST cluster in ST raster sets is the core in all locations, i.e., a subarea in which dynamic activity is recorded at various time stamps [36–38].

ST RASTER CLUSTERING

The grouping of ST raster sets that record measurements performed at different locations and times is a case in point in terms of a cluster of ST raster clusters. These clusterings are useful to find in many fields of industries. In the study of climate change, for example, it is useful to identify models that exhibit different climates on a global scale, compared to other models, and the rejection of similar models takes place to obtain the final list of unique climate models. The stacker data grouping is performed by comparing their representation networks, as mentioned in the above section.

### 4.3. Predictive Learning

In predictive learning, the goal is to learn "mapping" using an appropriate dataset, so that dependent variables/output values can be rendered through the independent variables/input values. In both dependent and independent variables, there is no binding regarding of the ST data type. Consequently, input values may have a different type from output values and this gives great flexibility in predictive learning methods; however, there are various problems when predicting learning when the input and output types are not the same.

When the input variable is a:

TIME SERIES

Using a time string in a location from a set of ST raster data as input to a predictive learning system is very common in sorting and regression problems. For example, time series, produced by the alternation of volume over time, are used to identify human speech by word sorting; this is performed with the proper use of neural networks [39]. With this method, modeling of input time values is achieved; however, they do not take into account the spatial properties that connect a time series [28]. This is solved using neural networks, which accept time and spatial characteristics as time series. An example of this method is

the traffic forecast study, taking both spatial and temporal characteristics as input values, which they receive from appropriate sensors. Thus, a prediction method extracts time series data from orbit data from which a prediction is derived by studying older paths of a moving object.

SPATIAL MAPS

Predictive learning can also result from the use of spatial maps, from a point in time of an ST raster or their subareas, as input values into a system. This method is often used when classifying images or identifying objects, using the deep convergent neural networks (CNNs) [40], which are widely used to input spatial data [39]. That is, relations are produced between the continuous spatial maps over time, with generalization and forecast results [41].

ST RASTER

Predictive learning can occur by using an entire ST raster as input to a system to export a dependent variable (output value). This method is used to study raster data from an FMRI test to determine if a patient has a psychiatric illness, as well as to identify any human brain activity, for example, the process of memorization [34,42]. This method is based on the representation of the ST raster, as a vector S × T, where S is a location and T is its time stamp, classifying, e.g., linearly, the incoming vectors resulting from a cluster of ST rasters. The method of linear alignment of vectors, however, is not always possible, because each grid ST raster is not always perfectly aligned with the others, resulting in heterogeneity between their spatiotemporal notations. An example of this heterogeneity is observed during the measurements in an FMRI examination, when the examined patient is in complete calm, where the time points cannot be grouped with other patients. An additional hurdle in applying the FMRI measurement method is that there are hundreds of cases, due to the thousands of brain regions being studied, per hundreds of time stamps. This results in the appearance of the "hyper-adjustment" model. This problem can be addressed by filtering the attributes output from each raster. There are predictive learning methods based on the use of tensors as input values, where a reduction in complexity is achieved based on the spatiotemporal interdependencies of the input data [17].

ST REFERENCE POINT

Prediction of a final situation in an area at a specific time can be carried out using predictive learning with input values as reference points. Usually, this can be achieved by studying neighboring locations and times. This applies to the prediction of ecological changes in an area, studying the changes that have occurred in neighboring and at specific times, as well as the prediction for the classification of an area in terms of its tendency to be included in a land cover category. This is similar to the spread of an infectious disease, studying the point of its occurrence at a given time and its neighboring spatial and temporal locations. The prediction can be made either by studying the spatial neighboring positions or the temporal ones. When using time information, the conditions are studied in older time markings. For example, the temperature level that will prevail in an area in a given month is derived from the temperature study of the immediately preceding months. Various methods are used to derive time series prediction, such as ARIMA models and space state models with exponential smoothing techniques, but there are also time series prediction methods that aim to detect cases that deviate from the set, such as hidden Markov models and Kalman filters. However, the spatial information in the above models is not taken into account in relation to time, something that needs improvement. Accordingly, prediction can also be made using spatial information, which is based on the correlation that exists in the measurements in relation to the sites adjacent to the study site. This is achieved using SAR models and GWR models [43], as well as the extraction of exceptions, using the Markov method. Today, an attempt is made to draw conclusions with the correlation of both space and time and the production of forecasting models [18].

### 4.4. Frequency of Pattern Mining and Display of Patterns

By studying spatiotemporal data, we can use various methods to export patterns. Patterns can be extracted, through the study of ST point type data, by observing common points in the dataset and exporting patterns, by observing common events occurring in adjacent spatial and temporal locations. For example, the goal of a study could be to identify two different events that occur close both temporally and spatially, such as the location of alcohol consumption and road accidents. With this method, event patterns are exported, representing two or more events that appear together. In the past decades, there has been extensive study of the occurrence of events in relation to space. Attempts are now being made to export co-occurrence patterns, taking into account both the space and the time of occurrence of these events. The export of successive patterns in ST points is also interesting. In the case of sequential patterns, an event type ST point can cause a series to appear from another type of case. For example, the occurrence of alcohol consumption can lead to the occurrence of a car accident and this can in turn lead to the occurrence of traffic jams. The purpose is to detect the interaction of one type of event with other types of events. The challenge today is to identify patterns that lead to productive conclusions about the sequence of events [44].

However, sequential patterns can also be extracted from track type data. These patterns contain areas that contain rolling of many moving objects moving in the same order. An example of a sequential orbital pattern is the routes followed by tourists moving in one area, from one popular attraction to another. The orbits, however, most of the time are not completely similar, so those that have many similarities should be included in the pattern. One method to accomplish this was developed by Tsoukatos and Gunopulos [45], where a series of ordered spatial points were used as rectangular areas to study successive patterns. Therefore, a trajectory follows a pattern if each part of it belongs to the rectangle of an area.

A similar method of pattern mining is when identified pattern areas in which similar sections of moving bodies orbit are observed. An important aspect in the study of sequential pattern of tracks is the correlation of the duration, during which the routes are made both from the starting point to the destination and also from point to point of the track. Another time criterion for exporting consecutive orbital patterns is when there is a small time difference in terms of visiting moving objects in the same locations following the same path. These patterns are called "herd" or "escort" patterns.

Patterns can also be extracted from time series. From a raster, time series patterns are extracted, which have repeated measurements at defined intervals, in various locations. For example, the cultivation cycle of a plant on a farm is a set of time series, a set of placements on the estate, following a pattern. To develop time series patterns, "matrix profiles" are constructed using vectors, which are the distance between time series sequences [11]. This method creates a time series subroutine table, based on the Fourier transform. The similarity between the time series subsets leads to the export of patterns. The ST time series exported from nearby areas are very similar to each other, resulting in easy export of patterns, also known as "time series consistency patterns". The time series coherence wall, for example, is the patterns extracted from series describing drought in neighboring areas in ST climate data. At the same time, there are the "evolving patterns ST", during which, in the spatial points of a location, a gradual evolution in the activity is observed that is studied over time. Methods such as tensor-based factorization, multitasking, and core time density estimation are now the focus of time series pattern processing and analysis, but they also deal with major problems mainly related to pattern heterogeneity [17].

Patterns, however, can also be extracted from networks that represent ST rasters. A network consists of nodes which interact with each other, and edges that determine the relationships of the nodes. From an ST raster network, patterns can be extracted as parts of the network that follow the same form of connectivity and are repeated at multiple points within the network. A special application in the export of network patterns is made during the study of ST raster data by using FMRI examination. Another example is the export

of climate change patterns observed in remote areas as different clusters of ST rasters, showing similar climate activity at specific intervals. However, the study of remote areas for the extraction of patterns can lead to erroneous results due to the spatial correlation that takes place, so it would be wise to study neighboring areas in an ST raster network where there is great spatial interdependence.

In addition, useful conclusions can be drawn from the observation of the anomalies presented by the ST data.

### 4.5. Detection of Abnormalities

Anomalies are defined as those cases which deviate from the set. The anomalies, or otherwise the extreme values in the study of data ST, identify some cases in the whole from which very interesting conclusions can be drawn. The identification of anomalies can be carried out during grouping, where cases are identified that cannot be included in a group due to the heterogeneity they present compared to the whole. Two methods of detecting anomalies have been developed, one being based on time and the other on space. When detecting anomalies by studying time series, "disagreements" are identified, while based on spatial data, "spatial extremes" are identified as cases that show heterogeneity, in relation to cases that belong to neighboring areas. However, having as a comparison both the time and the space for the detection of anomalies causes particular problems to appear. Thus, spatiotemporal data, which show a heterogeneity compared to the neighboring points that belong in a neighborhood, are easily identifiable endpoints. Initially, in this method, the ST points are first grouped using, e.g., St-DBSCAN, and then we identify the points that cannot be included in a cluster [26]. Another approach is an initial general grouping of points with extreme values detected, then detailed grouping by exporting extremes, and a final comparison of the extremes, between the first and second groupings, by final exporting of extremes.

Abnormalities are also detected during the study of time series. An orbit can be characterized as an orbit that is spatially located away from other orbits. Another approach to detecting track irregularities is to group the tracks based on their direction in an area, and locate that trajectory that cannot be integrated into a track grid [46]. Another method is to treat the trajectories as sets of nodes representing the regions, and edges as the paths between the regions, forming graphs. An extreme value considered in this case is the vapor heterogeneity of the graph [47]. It can also be considered a trajectory or group of trajectories, that shows different behavior in relation to its neighbors, for example, the movement of a group of people from region to region [41].

Anomalies are also detected during the study of ST rasters. Detection of group anomalies is performed in ST raster sets. Groups of neighboring areas represent an expected activity over a period of time. An example is a sudden and unexpected climate change observed in a group of neighboring areas, such as the occurrence of drought. To detect anomalies in ST rasters, extreme values are detected separately, studying the space and then taking into account the time. For example, the neighboring areas, which show different activity at each time point, are first identified and then grouped into sets with approximate time points that will form the ST raster groups with extreme values, as a step of post-processing the initial spatial sets. Another approach is to detect anomalies in time series data, which are then grouped based on space. However, there are approaches that use both time and space as criteria for identifying anomalies. An example is the detection of anomalies in the analysis of a video with repetitive scenes, where the difficulty here is to determine what the normal activity of a joint will be and to isolate those scenes that are different, which could also be repetitive and complex. This problem is addressed by defining dynamic texture model sets to determine the regularity, in terms of activity, then spatiotemporal maps of anomalies are drawn, in various areas, which are finally integrated through a filter of a random frame. Another method is to model the activity of the regularity of the scenes in space-time as a compact whole, identifying the fluctuations that appear in space-time as anomalies. There are also groups of anomalies in ST rasters that are located

at short intervals. An example is the tweets that mention the word earthquake in a short period of time in specific neighboring areas. These extreme values are called spatiotemporal miners [21,48–52].

### 4.6. Identifying Time Points of Change in Behavior of Object to Study

Detection of change determines the time point at which a change in behavior is observed in a study object. Change detection is studied extensively in time series. First, the parts that show similar properties at different time intervals are identified, and then those that deviate from them, identifying their time points of occurrence as time points of change. An example is the detection of periods of time during which a person experiences apnea while in a state of sleep [40]. There are, however, changes that occur periodically, known as "periodic" time series. With regard to exporting changes to ST raster data, in addition to time, space must also be taken into account. For example, change is a time series that differs from others in a spatial context, also known as "contextual", with a spatial context, to define the area in which similar time series represent a time period, or neighborhood (nearby areas) of a set of time series.

Another attempt at detecting changes is to locate both time and space during which a change in an ST raster is observed, i.e., time intervals in specific spatial contexts during which a change in the behavior of a system takes place, for example, the detection of areas where, at some point, vegetation loss is observed due to deforestation. The detection of spatiotemporal changes is also performed during the video analysis for the detection of scenes (as areas) that deviate from the total of the scenes in some time intervals. Another approach is to define an area as a point in the form of a grid or line, and the time interval at which the change takes place as a point in time.

Having identified the similarities and differences between the ST data, important relationships between them can now emerge.

### 4.7. Relationship Development

Relationship between time series are extracted based on the criteria taken into account to determine similarity between them, but when you take these into account in addition to time and space then there are various ways to determine the relationships between ST data. A form of relationship can be drawn between remote groups belonging to neighboring areas by comparing them. This approach finds a wide field of action both in the study of brain activity and in the study of climate change. The purpose is to distinguish between the relationships of sets of sites and the relationships that exist per two locations, belonging to the above sets of sites. One approach uses tensors to extract regions and relationships during the FMRI brain scan [17,41,45,53]. Research has been carried out where a network is derived from FMRI data when a healthy person is at rest, where it is observed that there is a high similarity to networks derived from FMRI data obtained when examining other healthy people who are also at rest [42].

However, the relationships that arise between ST data do not have a constant force over time. For this, the time window in which a relationship between two areas is determined should be strictly defined. One method is to derive patterns, in terms of dynamics, of time series relationships from different spatial contexts that are applicable to ST FMRI test data to identify remote areas by correlating them.

Another method is by using graphs, whereby, for a region, a graph is constructed at a time, and the relation is extracted from that graph. When exporting relationships, delayed relationships are also found in time series; these are relationships between two areas affecting each other, changing the situation for each other, with a time delay. When relationships that are delayed take a while, then we are talking about dynamic relationships that are delayed. In dynamic delay relationships, the goal is to identify the time intervals in which two sets have similarities.

Relationships delays in time series indicate the existence of a cause that causes the change between two areas. This can lead to the prediction of a dependent variable, keeping

in mind the cause/input that causes the change in a situation. However, there can also be a causal relationship between causes that lead to a situation. The existence of these relationships led to the design of Granger causal maps for time series, which depict many variables with the causal relationships between them. Granger's application of causal maps was performed in climate science, where the purpose was to determine graphic models from causal edges between nodes.

Thus, having all these features and properties, ST data are used in many applications today. We present some of the applications for which ST data are used nowadays.

## 5. The Application of St Data in Various Fields Today

In many fields today there are very large volumes of data that are difficult to be manage, so computer science is called upon to deal with this challenge in order to obtain new valuable knowledge, such as in the fields of medicine, social media, and meteorology [1,36]. However, in order to extract accurate results from these data, both the parameter of space and the time of their reception must be taken into account, so we are now talking about an attempt to process space-time data. However, both their source and application for which they are intended play a role in their collection. The sources of ST data, and also their analysis and application, come from various fields, such as medicine (healthcare and epidemiology), meteorology (environmental quality and climate change), social media, agricultural development study, traffic congestion, criminology, and, finally, the study of the solar system, focusing on the study of the sun [3].

More specifically:

In medicine, in the field of the study of brain activity, through neuroimaging, spatiotemporal data are used for analysis to extract patterns [5,6]. With FMRI (functional magnetic resonance imaging) [53], EEG (electroencephalogram), and MEG (magnatum), the required data are obtained to record human brain activity. Through their analysis, valuable patterns of disturbed human behavior can be extracted, and patterns of brain activity corresponding to mental disorders can be identified. Each test follows a different data acquisition method; for example, FMRI scans the brain, trying to locate, in millions of areas, the areas where there is neural stimulation during an activity. In addition, the EEG (electroencephalography) method works the same, with the only difference that the scanning is performed only in dozens of areas of the brain. In addition to the area, there is a difference in the time of data download in the above methods. The FMRI receives data every 2 s, while the EEG every 1 ms [54]. The combination of data obtained from the above data acquisition methods is the final form of data that is analyzed in order to map the human brain in specific activities and to extract the required standards. Another medical field that uses ST data is epidemiology [49]. In hospital records, the medical history of patients, patient data, medical opinions, and suggested modes of treatment are recorded in different regions of the world and at different time points. These data are perhaps one of the most representative examples of spatiotemporal data, since at the time of their capture, both the place and the time of their receipt are recorded. The analysis of these data aims to extract patterns of patients and diseases in terms of the occurrence of specific diseases, such as influenza, cancer, and COVID-19. At the same time, other valuable conclusions can be drawn about the disease, such as the extent of its spread. However, in combination with other data, such as climatic conditions and environmental characteristics, other factors could be found that favor the existence and spread of a disease, providing valuable knowledge that could protect a large number of people.

Respectively, extraction and analysis of spatiotemporal data are carried out during the study of both climate change and during the study of environmental pollution. Specifically, for the study of climate change, data are collected that refer to measurements of temperature, humidity, wind intensity, and atmospheric pressure in different areas and at different times. Measurements are performed both by methods on the ground and by the use of satellites. The analysis of these data is aimed at extracting patterns in relation to climate change, providing knowledge that can help people in predicting difficult weather conditions and

taking timely protection measures. Conclusions regarding the quality of the atmosphere and the aquifer can be drawn by analyzing measurements related to the percentages of air content of carbon dioxide, nitrogen dioxide, and others, and for water conductivity and pH level, respectively. These measurements are usually made with sensors both in water and in the air. Another factor that affects the quality of the environment is noise pollution, which is also measured with similar sensors. The analysis of all the above measurements and data leads to the extraction of the level of environmental pollution while giving the possibility of drawing conclusions regarding the cause of pollution, providing knowledge that can lead to proper environmental protection measures. Measurements and data related to climate change and environmental pollution are used in combination with measurements in terms of the size of large geographically cultivated areas, via satellite, with measurements related to fertilization, weeds, and planting of plants in these areas. With proper analysis and elaboration, we can draw valuable conclusions for agriculture, for example, in terms of defining measures to make agriculture more efficient [55].

In the light of the extraction, study, and final analysis of ST data, the study of the sun also moves. The various changes of the sun in terms of its form and composition affect the state of the whole solar system. The observation of the sun is performed on a daily basis, with measurements concerning the flame, its active area, the sunspot, etc. These measurements in relation to the time at which they were made constitute the set of spatiotemporal data. The analysis of these data leads to the export of patterns, related to events that take place in the solar system, but patterns are also exported in relation to the climate change observed on Earth [49].

A completely different field in terms of the above that also deals with the analysis of ST data is criminology. When studying a crime, data such as the nature of the crime and the time and place where it was committed are taken. The analysis aims to extract patterns of crimes based on their nature, and also patterns based on the frequency and area in which they occur. This knowledge helps to formulate a sound policy to reduce crime.

However, social media is an important source of data. Each user's post informs us about the time, place, and feeling they experienced while recording an experience [3]. Thus, by analyzing all these ST data, which is enriched daily, a user's profile can be extracted with great accuracy, and one can watch the spread of an infectious disease or the impact that some political currents have. These data are valuable for drawing conclusions about a society in almost all areas.

In the study of human activities, interesting conclusions can be drawn between the data obtained and the time and the area of embarkation and disembarkation, as well as the route. The analysis of these data can show the normal movements of a population, even compared to the weather conditions that prevail at a specific time, in an area. Patterns can still appear regarding the routes followed by professional taxi drivers. The study in this field can allow the design of better highways with the aim of avoiding traffic congestion and also the choice of the best possible route depending on the time and the area.

A big problem today is energy saving, using the least possible fuel consumption for means of transport, both for ecological and economic reasons. Studies are being carried out today regarding the calculation of fuel consumption as well as predictions of different routes that vehicles follow for better fuel savings. However, these proposed studies are limited both in the distance that a vehicle is going to follow (they are mainly applied to short distances) and in the driving preference that the driver will have. Consequently, these methods cannot be applied, firstly, over long distances, and secondly, due to the nature of the road network. Within an urban area, a new method, PLd-FeRR, is available today to solve this problem. It first takes into account the driving behavior of the driver, based on the history of the routes they have taken, and then identifies the possible factors that may lead to high fuel consumption over long distances. Individual characteristics that accompany a type of driver, such as the amount of fuel consumed over long distances, are entered into the PLd-FeRR model and a prediction is made for the best possible route with the least possible fuel consumption [56].

For a large urban area, the heavy traffic congestion in various parts of it is a problem. Prediction through space-time mining provides a solution to this. Methods that base prediction on using deep learning are unable to cope with the prediction needs due to the large amount of training data they need, which is not currently readily available. Today, an effort is being made for this knowledge to come from multimodal spatiotemporal data of a city or road links of cities. The ST-DAAN model transfers the desired knowledge to the network for prediction extraction. With this model, prediction of congestion points within a city or at points between cities is carried out to a very satisfactory degree [56].

The STUGN spatiotemporal sampling graph collection network meets the needs for daily traffic forecasting in a city. STUGN reflects an urban road network with virtual nodes, with the ability to capture local as well as global correlations between the spatiotemporal data collected. A time graph is created in which the time correlations between neighboring time points, both locally and globally, are captured [57].

Due to the dynamic traffic data, which is constantly changing over time, a new deep learning model, the DTC-STGCN spatiotemporal graph convolutional network, has been proposed for traffic and speed prediction with dynamic ST dependencies. This model outputs a matrix representing the dynamic ST correlations of neighboring moving objects. The GCN model captures the characteristics of the neighboring dynamic spatial data of an urban road network, while the LSTM (long short-term memory network) model captures the characteristics of the temporal data of the road network. The GCN model combined with the LSTM creates a hybrid model, representing dynamic spatial and temporal correlations for traffic congestion prediction with very high performance rates [58].

Today, efforts are being made by researchers in order to utilize spatiotemporal data in ways such as evolving existing systems or developing new algorithms that process large volumes of spatiotemporal data (Figure 2). The current scientific fields where researchers can expand their research are of four types: spatial databases, methods for processing large volumes of spatial data, programming languages, and, finally, GIS software. This effort of the researchers aims to create powerful mining systems both for spatial and spatiotemporal data [2].

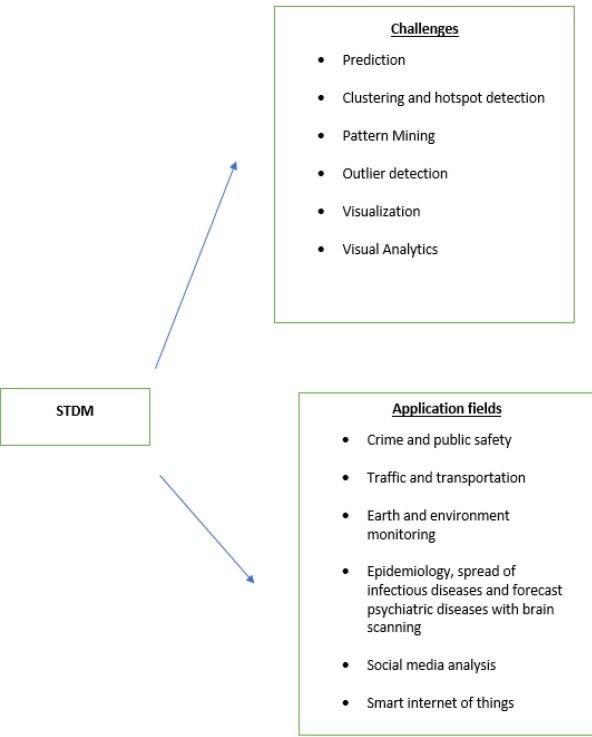

**Figure 2.** Summary of challenges related to ST data and the applications they are expected to serve.

## 6. Future Work

In order to solve difficult and complex problems in the future, new STDM methods should be developed that can combine and accept, as a dataset, data of various types coming from various sources. Future research is focused on modeling and visualization of various combinatorial STDM methods so that complex scenarios of various situations can be solved. The large amount of geographic data, combined with its mining timestamp, can provide great answers to complex problems. The relationships and patterns that can be extracted through spatiotemporal data mining are the cornerstones for analyzing and deriving useful conclusions, for studying a situation and solving complex problems through various applications. Thus, researchers have focused their interest on the design of algorithms for the best grouping and modeling of spatiotemporal data for the extraction of appropriate relationships and patterns, which will lead to the extraction of safe predictions. An important challenge is the complexity of the relationships of spatiotemporal data types. The spatiotemporal data, in order to be used to extract solutions to complex problems, come from various scientific fields with a wide variety of different interdisciplinary characteristics, which leads to complex relationships between them. For this reason, it is advisable in the future to develop the appropriate algorithms that will be able to unify and combine interdisciplinary characteristics as a data of spatiotemporal data with purpose the extraction of appropriate relationships between them. This will lead to clustering, modeling and pattern extraction, which will now respond and adapt to extensions and variability of data, due to their direct dependence on time.

## 7. Conclusions

In recent decades, a great deal of research and study has been carried out regarding the extraction of spatiotemporal data. This article is a very good introduction to the study of different STDM methods. We also present the types of data ST, the conversion of one type to another, the problems and challenges they are called upon to answer, the methods, patterns, groupings, the detection of anomalies, the changes, and the relationships that they present.

The challenge today is to identify new representations of stack data. The representation and study of spatiotemporal data, in combination with other properties that accompany all natural phenomena, are currently a challenge in various scientific fields.

**Author Contributions:** Conceptualization, E.K.; Validation, E.K.; Writing—Original Draft Preparation, E.K.; Writing—Review & Editing, E.K.; Visualization, G.V.; Supervision, N.P. All authors have read and agreed to the published version of the manuscript.

**Funding:** This research received no external funding.

**Institutional Review Board Statement:** Not applicable.

**Informed Consent Statement:** Informed consent was obtained from all persons involved in the study.

**Data Availability Statement:** Not applicable.

**Conflicts of Interest:** The authors declare no conflict of interest.

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
