# Peer review of "Spatiotemporal Data Mining Problems and Methods"

_2813-2203, doi:10.3390/analytics2020027_

Round 1
Reviewer 1 Report
This paper provides a survey on the research problems and methods in the field of patio-temporal data mining. Indeed patio-temporal data mining is a increasingly important research problem and has attracted more and more research attention in both academic and industry. However, I have some concerns on this papers. Detailed comments are as follows.
1) There are many survey papers on STDM, including both traditional methods and deep learning methods. How this work is different from existing surveys is not clearly explained. What is the new contributions of this paper?
2) The logic of the paper is not clear. The paper should be carefully re-organized. The second section is application, which is quite strange to me. Why the authors introduce application first? In the second section, the authors focus on introducing ST data. However, in section 3.5 they introduce similarity between data instances, which makes me confused. The logic and organization of section 3 is not clear and not focused.
3) In section 4, it is better to clarify what is difference and unique challenges of these problems between ST data and other types of data. For other types of data, there are also similar problems such as clustering, frequency pattern mining and prediction.
4) What is the challenges in the area of STDM and what are the potential future research directions are not discussed in the paper.
5) A lot of relevant works are missing to cite and do not discussed in the paper. In section 2, more relevant works should be cited as there are so many applications for STDM. The following works are suggested to cite and discuss in the paper.
Spatio-Temporal Knowledge Transfer for Urban Crowd Flow Prediction via Deep Attentive Adaptation Networks. IEEE Trans. Intell. Transp. Syst. 23(5): 4695-4705 (2022)
Deep Learning for Spatio-Temporal Data Mining: A Survey. IEEE Trans. Knowl. Data Eng.34(8): 3681-3700 (2022)
Personalized Long-distance Fuel-efficient Route Recommendation Through Historical Trajectories Mining. WSDM 2022: 1072-1080
- Dynamic graph convolutional network for long-term traffic flow prediction with reinforcement learning. Inf. Sci. 578: 401-416 (2021)
- Deep Irregular Convolutional Residual LSTM for Urban Traffic Passenger Flows Prediction. IEEE Trans. Intell. Transp. Syst. 21(3): 972-985 (2020)
- 6) There are many typos and grammar errors in the paper. A careful proofread is needed. For example, the first word in a sentence should be capitalized, but in many parts of the paper (section 3.2 especially), they are not capitalized.
Reviewer 2 Report
Summary:
This work tries to present a comprehensive survey in spatial-temporal data mining and analysis. However, it fails to do so due to 1) unclear novelty comparing to existing works; 2) poor readability brought by unprofessional writing and lack of logical organization; 3) too broad topic instead of focusing on certain aspect in ST data mining.
Strengths:
1.This is a comprehensive work covering ST data applications in different domains.
Weaknesses:
1.The writing needs to be significantly improved. This reviewer found quite a few issues from the abstract alone. Please proofread the whole submission carefully to examine similar issues.
2.The novelty of this paper is unclear.
3.The paper needs major revision regarding presentation and organization.
Comments:
1.Why are some of the words not capitalized in the title?
2.line 11-12, "do not take into account the data acquisition time their extraction", grammar mistake?
3.line 16, "it's", should avoid use such contraction.
4.line 23, "st" should be defined to avoid confusion.
5.line 26, "Instances", why is this capitalized?
6.line 26, better avoid using "problems / challenges”? maybe use “and”
7.The abstract does not mention the novel contribution of this paper, is this the first paper summarizing spatial-temporal (ST) data mining and analysis? If not, how is this one different from the existing one.
8.Also, the topic seems pretty broad according to the abstract.
9.line 56-67, too many similar examples to demonstrate the same point.
10.line 53, it talks about need for new methods. But the rest of the paragraph barely summaries the methods but goes into different cases.
11.line 78, “stdm” should be “STDM”, in consistency with the above. This reviewer will not point out all such small issues. Please check by yourself.
12.The introduction is too lengthy, it should mention line 90 in an earlier place, i.e., what is the current problem and what does this paper try to address.
13.Introduction section misses novelty summary.
14.Section 2, for survey type paper, it would be nice to have a table or graph to summarize the applications/data types.
15.line 118, “I mean EEG works “, too verbal.
16.Section 2, these applications lack summary to tie them closely back to the problem this paper tries to address.
17.Section 3.1, there is no reference for this part. Also, what is the purpose of this section?
18.”This therefore leads to the need to 199 develop algorithms for data mining, which take into account both space and time. “ this is little information in such sentences. And this happens a lot.
19.Section 3.2.1, please use proper subscript format to increase readability.
20.Section 3.4.2, 3.4.3, 3.4.4, these concepts are not independent. Better categorization is needed.
21. Section 4, it needs a coherent framework to introduce such diverse applications. Otherwise, readers are easily lost.
Round 2
Reviewer 1 Report
The authors have well addressed my previous concerns. I suggest to accept it as is.
Author Response
Thank you for your patience and reviewing work.
Reviewer 2 Report
Weaknesses:
1.The writing needs to be improved. Some of the writing fails to concisely convey the idea to the readers. Instead, it uses general expressions rather than efficiently summarizing the work.
2.If the paper is designed to be a survey, it should have some table summarizing the related works to give readers a clear idea of how they can leverage this paper.
Comments on cover letter:
1.It is better to attach details to the cover letter so reviewers don’t have to guess or look through the paper to see what revisions have been made, e.g., it is unclear what changes are made by reading a response like “The paper has been reorganized. “
2.The comments are not well addressed. There is no one-to-one responses to some of the comments left by the reviewer. E.g., comments 1 to 7 are not replied and comment 9 is replied with irrelevant response.
3.One key question remained unanswered: what are the takeaways can only be learned from this paper? “The purpose of this article is to provide a good introduction to spatiotemporal data” is not enough. This can be learned from other papers as well. Valuable learning can be summary of limitations of existing studies, thoughts on overcoming them, etc.
Comments on revised manuscript:
1.It is a bit uncommon to see a detailed summary of each section in the abstract. It saves the efforts for the authors but not in an elegant way. It would be better to shorten such summary by a concise and clear story with motivation, approach, and results.
2.line 24, any reason choosing to introduce these methods/problems? It would be better to separate problems and problems instead of dumping them in one sentence.
3.line 27-28, such sentences are barely informative since it does not tell the readers what conclusion.
4.table 1, what attributes are being compared here, missing one column on the very left to denote them?
5.line 48, “toco”?
6.Section 2, how is this work different from the ones mentioned? The novelty is still not summarized in the revision. The authors mentioned “Description Logic” to address the novelty comment, but it is only used in few sections, not enough to be considered as the main novelty.
7.Figure 1, a bit confusing. Maybe just combine a and b to indicate trajectories with ST data. Besides, it seems unnecessary to have a figure for it since little information is contained.
8.line 210, please use math notation for variables such as “li” and “ti”.
9.Figure 2, what is the purpose of Figure 2?
10.line 159, how can one leverage the “limitations” and “rules” summarized for different data types?
11.line 840, there is few reference for this section, which is inappropriate for a survey paper.
12.line 905, this section should be better organized, maybe starting with what applications will be covered and introduce one by one. One would expect the authors will introduce medicine, meteorology, etc. after “The sources of ST data are specific: medicine (healthcare and epidemiol- 913 ogy), meteorology (environmental quality and climate change), social media, agricultural 914 development study, traffic congestion, criminology and finally study of the solar system 915 focusing on the study of the sun.”, but it did not.
Author Response
Thank you for all your patience and detailed comments which really help to improve the paper a lot. Please check the attachment.

Round 3
Reviewer 2 Report
1.line 20-22, some expressions are too verbal, e.g., "something that we hope..." 2.line 853, two periods.